# Translational Gap between Guidelines and Clinical Medicine: The Viewpoint of Italian General Practitioners in the Management of IBS

**DOI:** 10.3390/jcm11133861

**Published:** 2022-07-03

**Authors:** Massimo Bellini, Cesare Tosetti, Francesco Rettura, Riccardo Morganti, Christian Lambiase, Gabrio Bassotti, Pierfrancesco Visaggi, Andrea Pancetti, Edoardo Benedetto, Nicola de Bortoli, Paolo Usai-Satta, Rudi De Bastiani

**Affiliations:** 1Gastrointestinal Unit, Department of Translational Sciences and New Technologies in Medicine and Surgery, University of Pisa, 56010 Pisa, Italy; mbellini@med.unipi.it (M.B.); christian.lambiase93@gmail.com (C.L.); pierfrancesco.visaggi@gmail.com (P.V.); andreapancetti10@gmail.com (A.P.); nicola.debortoli@unipi.it (N.d.B.); 2Department of Primary Care, National Health Service, 40046 Bologna, Italy; tosetti@libero.it; 3Italian Group for Primary Care Gastroenterology, 32032 Belluno, Italy; edoardo4@gmail.com (E.B.); rudeba@libero.it (R.D.B.); 4Clinical Trial Statistical Support Unit, Azienda Ospedaliero Universitaria Pisana, 56010 Pisa, Italy; r.morganti@ao-pisa.toscana.it; 5Gastroenterology and Hepatology Section, Department of Medicine, University of Perugia, 06516 Perugia, Italy; gabassot@tin.it; 6Gastrointestinal Unit, National Health Service, 87100 Cosenza, Italy; 7Gastroenterology Unit, Brotzu Hospital, 09121 Cagliari, Italy; paolousai@aob.it; 8Department of Primary Care, National Health Service, 32032 Belluno, Italy

**Keywords:** irritable bowel syndrome, general practitioners, Rome Criteria IV, Bristol Scale, primary care

## Abstract

Irritable bowel syndrome (IBS) guidelines are generally developed by experts, with the possibility of a translational gap in clinical medicine. The aim of our study was to assess an Italian group of general practitioners (GPs) for their awareness and use of criteria for the diagnosis and management of IBS. For this purpose, a survey was carried out involving 235 GPs, divided into two groups according to their years of activity: 65 “junior general practitioners” (JGPs) (≤10 years) and 170 “senior general practitioners” (SGPs) (>10 years). JGPs were more familiar with the Rome IV Criteria and Bristol Scale than SGPs. Abdominal pain, bowel movement frequency and bloating were the symptoms most frequently used to make a diagnosis. The most probable causes of IBS were reported to be abnormal gastrointestinal motility and psychological triggers. SGPs reported more frequently than JGPs that challenging management and patient’s request were motivations for a gastroenterological consultation. The practice of clinical medicine is still far from the guidelines provided by the specialists. Abdominal pain related to defecation and changes in bowel frequency are considered to be the more important symptoms for IBS diagnosis, but most GPs, both JGPs and SGPs, like to consider abdominal bloating as another useful symptom. Involving both gastroenterologists and GPs in developing shared guidelines would be highly desirable in order to improve IBS management strategies in everyday clinical practice.

## 1. Introduction

Irritable bowel syndrome (IBS) is one of the greatest clinical challenges for both gastroenterologists and general practitioners (GPs) [1,2,3,4,5]. The complex pathophysiology of IBS often leads to an iteration of medical examinations and diagnostic tests, and to a multiplicity of different, and sometimes contradictory, therapeutic indications. The inevitable result is often an unsatisfactory management of patients complaining of IBS symptoms [6,7]. The definition of IBS is still under debate, as well as the commonly suggested diagnostic criteria, such as the Rome IV Criteria (RC) shown in Table 1 [8]. These are scarcely used in medical practice, even by gastroenterologists, although they continue to be a milestone for scientific research [9]. Furthermore, given that most IBS guidelines have been conceived by gastroenterologists, it is likely that there is a translational “gap” with GPs, and thus some difficulty occurs in both their diffusion and acceptance in primary care [10]. This is particularly important in the Italian context, where most GPs have several years of activity and the generational change among GPs is taking place slowly. This could mean that Italian GPs are not sufficiently up-to-date, especially given the absence of decisive and substantial changes in the management of IBS guidelines in the last ten years. Conversely, “junior GPs” (JGPs), i.e., those with ≤ 10 years of practice, who have a more recent university and educational background and are also in direct liaison with specialists, could have more up-to-date knowledge and may be able to integrate these concepts into their medical practice.

The aim of our survey was to assess the awareness and the use of criteria for the diagnosis and management of IBS in a cohort of Italian GPs.

## 2. Materials and Methods

The study involved GPs randomly selected from those working in three cities in the three regions of northern (Lombardy, Veneto, Emilia-Romagna), central (Tuscany, Marche, Lazio) and southern (Campania, Calabria, Sicily) Italy. An invitation with a link to an anonymous online form was sent in October 2019 to 400 GPs. The form was, fundamentally, an example of simple data collection, based on information obtained in previous studies on GPs, and therefore did not need validation [10,11]. The survey answers contained demographic information (e.g., age, gender), and stated whether the participants were specialized in gastroenterology. The participants were divided into two groups according to their years of activity: those with ≤10 years of activity were JGPs and those with >10 years were “senior GPs” (SGPs). They were also asked to rate whether their IBS knowledge was satisfactory or unsatisfactory, and whether they needed specific updating on this topic. Finally, the GPs answered questions about their knowledge and clinical use of the RC IV and of the Bristol Scale (BS), and the essential symptoms for IBS diagnosis. Moreover, they were asked about the possible causes of IBS, the reasons for a referral to a gastroenterologist, and to rate the satisfactory management of their IBS patients. The relevance of the possible causes of IBS was rated as likely or unlikely, whereas satisfactory IBS management was defined as a patient who perceives their symptoms as no more than a nuisance and does not require further medical care [10]. The survey was totally anonymous and there was no possibility of identifying the participants; the GPs had only to declare their consent to participate. The study complied with the principles of the Helsinki Declaration, without needing the consent of an Ethics Committee.

### Statistical Methods

The SPSS v.26 technology was adopted for data analyses. Absolute and relative frequency and mean and standard deviation were used to describe categorical and continuous factors, respectively. Qualitative data were described by the z-test for two proportions or by or chi-square test, and quantitative data were compared by the *t*-test for independent samples (two tailed). Significance was set at 0.05. 

## 3. Results

Two hundred and thirty-five GPs (58.75%) were part of the survey. The main features of the participants, including significant differences, between SGPs and JGPs are shown in Table 2 and Table 3. The participants were mainly males with a mean age of 52.0 ± 14.6 years. JGPs (*n* = 65) were mostly female, in contrast to SGPs (*n* = 170) (67.8% vs. 30.6%; *p* = 0.001). With regard to educational needs, 52.8% of GPs felt that they had a satisfactory knowledge of IBS. Comparing GPs with and without a gastroenterology specialization, those in the first category reported that their IBS knowledge was satisfactory more frequently than those in the second (65.4% vs. 20.2%; *p* = 0.001). The percentage of GPs who considered professional training on IBS useful was 39.1%. No correlation was observed between years of activity and educational requirements. 

Table 4 shows the knowledge and use of the RC IV and the BS. Awareness of these tools was low (54.5% and 52.8%, respectively), but most of the GPs who stated that they were familiar with the criteria declared that they used them in clinical practice (75.8% and 74.2%, respectively). JGPs were more often familiar than the SGPs with both the RC IV (72.3% vs. 47.6%; *p* = 0.001) and the BS (69.2% vs. 46.5%; *p* = 0.003). GPs specializing in gastroenterology were more often familiar with the RC IV (76.9% vs. 40.6%; *p* = 0.001), than the GPs without specialization, but not with the BS. Among the GPs familiar with the RC IV, 56.3% said they would add other symptoms to the RC IV definition of IBS, particularly abdominal bloating (84.7%) and abdominal discomfort (43.1%).

Figure 1 shows that the most common symptoms used to diagnose IBS in clinical practice were abdominal pain related to defecation, changes in bowel movement frequency and abdominal bloating, with no differences between SGPs and the JGPs. Conversely, a minority of GPs mentioned difficult or incomplete defecation, defecatory urgency and emission of rectal mucus. There were no differences between GPs specialized in gastroenterology and their colleagues without specialization in using “abdominal pain related to defecation” to diagnose IBS.

Table 5 summarizes the frequency of the most “likely” causes regarding IBS pathophysiology. Abnormal gastrointestinal motility and psychological triggers were considered probable causes of IBS by more than 60% of GPS, whereas gut dysbiosis, visceral hypersensitivity, gastrointestinal infections, and food intolerance and/or allergy were perceived as “likely” hypotheses by less than 60% of the GPs, with no difference among SGPs and JGPs. 

The different reasons for gastroenterological consultations are shown in Figure 2. Failure of therapy, need for second level testing and patient reassurance were reported very frequently, with no significant differences between SGPs and JGPs. Satisfactory management of the patient was achieved by 49.3% of GPs.

## 4. Discussion

This study reports the viewpoints of a group of Italian GPs regarding the management of IBS. The prevalence of males in our survey, as well as the high mean age, reflects the Italian context, in which most GPs are males, and generational change takes place slowly. With regard to educational requirements, only 39.1% of GPs believe that an update on IBS would be useful. Indeed, more than half of them consider their knowledge of this topic to be satisfactory. In comparison with a previous survey carried out by our group [10], the rate of GPs considering their knowledge to be satisfactory is substantially the same, while the number of those who considered specific professional training helpful increased threefold. No correlation was observed between years of activity and the above variables, but gastroenterologists more frequently considered their IBS knowledge to be satisfactory.

Although the RC IV is considered fundamental in diagnosing IBS for research purposes, many studies have revealed the low frequency of their utilization among GPs [12,13]. A study conducted across Europe found that only 23% of GPs knew the RC, and only 20% used them in their clinical practice [14]. Furthermore, in our study, the GPs’ awareness of the RC IV for the diagnosis of IBS was low, although it had increased in comparison with the results of the study we conducted in 2005 (54.5% vs. 35.7%). Further, their use in current practice among those familiar with the RC IV has increased (75.8% vs. 60%) [10]. These data confirm that awareness and use of the RC IV are widespread only among the gastroenterologists involved in the management of functional digestive disorders [1,12,13,15,16]. This is in contrast with GPs, who, despite playing an important role in diagnosing and treating most IBS patients, scarcely know or use the RC in their clinical practice [12,17,18,19]. The positive diagnostic criteria for IBS are mostly based on experts’ opinions and not on high-quality evidence. As far as their applicability in primary care is concerned, there are very few data [20] or studies validating a positive diagnostic approach to IBS diagnosis. Begtrup et al. demonstrated that a positive approach was not inferior to an exclusion approach, with less use of healthcare resources and lower direct costs, and with similar effects on the clinical status of the patients [21]. However, currently, IBS diagnosis is still mainly an exclusion diagnosis for many GPs worldwide [13,22,23,24]. It is a laborious procedure that incurs high costs and requires a great deal of time [22]. JGPs are more familiar with the RC IV than SGPs, showing that knowledge of the RC IV has been steadily increasing in recent years.

In addition, regarding awareness of the BS, JGPs are more familiar with it than SGPs presumably because they consider the BS to be quick and easily comprehensible for patients due to its visual immediacy. Unsurprisingly, GPs specializing in gastroenterology are more familiar with the RC IV, but not with the BS, than other colleagues.

More than half the GPs who reported being familiar with RC IV believed that additional symptoms should be introduced to the IBS definition to improve the clinical picture of IBS patients. The percentage of GPs who would introduce abdominal bloating among the diagnostic criteria for IBS is 84.7%. In addition, 43.1% of GPs thought that abdominal discomfort, which was present in the RC III but was removed from the RC IV, is also an important symptom. Years of activity and gender did not seem to correlate with any choice of answer among the GPs.

Abdominal pain related to defecation, changes in bowel movement frequency and abdominal bloating, as reported in our previous survey [10], were the symptoms most frequently used in clinical practice by GPs. The prevalence of bloating is substantial, rising to 66–90% in IBS patients [25]. Abdominal bloating is reported to be the most annoying symptom, not only by patients but also by GPs [26], it being considered the third most frequent symptom of IBS by both GPs and gastroenterologists [12]. However, according to the RC IV diagnostic criteria, bloating is only noted as a supporting criterion [8]. Even in the NICE guidelines, bloating was only a supporting factor [27], but in the 2015 revision it became one of the main symptoms [28], and according to a European consensus created for the diagnosis of IBS in primary care, the distinguishing features are: altered bowel habits, bloating and abdominal pain [29]. Bloating within the clinical context of IBS is related to an increase in symptoms and pain, depression, fibromyalgia, somatization and alteration of dietary fluid composition. Clinical outcome could be improved if these factors were recognized and addressed during the diagnostic and management process of IBS patients [30].

Regarding pathophysiological mechanisms, the most “likely” causes were abnormal gastrointestinal motility and psychological triggers, in line with our previous survey [10], with no differences between SGPs and the JGPs in their answers. However, other causes (e.g., gastrointestinal dysbiosis and infections, visceral hypersensitivity and food intolerance/allergy) were also included as not infrequent pathogenetic mechanisms. The present study confirms the broad areas of uncertainty of GPs regarding IBS etiology, also reported by previous studies [31]. Psychological factors and stress are considered the most probable causes by a large number of GPs [9,13,14,22,31]; however, no consensus exists on the roles of other possible mechanisms [13,14,26]. Indeed, the pathophysiology of IBS is not fully clarified, and it is still the focus of intense debate, in which many different, sometimes contradictory, hypotheses have been suggested [6]. Interestingly, in this study, food intolerance/allergy was considered the least likely cause. In recent years, many studies have shown that IBS is strongly influenced by diet [32,33,34], so that food is regarded as a symptom-precipitating factor by many IBS patients [35,36]. This could be an indication of insufficient updating of GP knowledge, or at least an indication that they do not thoroughly investigate the patients’ medical histories.

Regarding the motivations for a gastroenterological consultation, the GPs indicated several reasons (i.e., failure of therapy, the need for a second level test, the need to reassure the patient, patient’s request and challenging management). The frequency of GPs referring patients to specialists was high, with reasons varying according to the different countries (presumably due to differences in the organization of the local National Health System) [5,13,14,37,38,39,40]. The frequency of referring IBS patients from primary health care to secondary health care services was 4%–32% [10,14,40]. Mujagic et al. noted that a referral to a specialist may be helpful in reassuring patients with symptoms partially responding to the GP’s treatment and/or in improving patients’ satisfaction [9]. In another study it was reported that male and older physicians referred patients to gastroenterological consultations more than female and younger physicians [41]. Similarly to a previous survey, no difference was observed in this survey regarding specialist referral according to doctors’ age and sex [10]. However, these data must be evaluated in the light of the assessment of satisfactory patient management achievable by about 50% of GPs. Compared to our previous investigation, this result is quite different (62.7%) [10]. In the literature, the need for a specialist visit as a result of patient’s request is reported to vary widely [10,42]. Mira et al. reported that IBS patients’ satisfaction with the care provided by their GP or specialist was comparable, although the same authors showed that patients would prefer to be addressed more often by a gastroenterologist, despite the reluctance of GPs [43]. Indeed, the need for specialist visits is a sensitive issue for GPs, arising from emotionally charged relationships with patients, colleagues, gastroenterologists and supervisors. As a result, the decision-making process is strongly influenced by environmental, personal and clinical conditions [9,44]. A limitation of our study could be the number of participants and the fact that those who agreed to participate were probably more sensitive to the topic, although this is a problem common to any internet survey. However, GPs more familiar with IBS were not the only participants selected for the survey; indeed, many GPs reported insufficient knowledge of IBS.

Finally, an additional limitation could be the lack of a control group (e.g., gastroenterologists or IBS experts). Nevertheless, this was beyond the scope of the present investigation, which focused only on GPs.

## 5. Conclusions

The study clearly shows that it is essential to constantly update GPs on IBS. A gap still exists between expert guidance, i.e., national and international guidelines, and clinical medicine. A probable reason for the gap persisting over the years is the inadequate dialogue between the various healthcare providers. Specialists do not completely convince GPs, while it is possible that the latter, on the other hand, do not adequately express the reality they face every day, along with their requirements and needs. It should be highlighted that, according to all guidelines [45,46], the diagnosis of IBS should be made through a positive approach, which would help to reduce the consumption of healthcare resources. Therefore, among the activities of GPs in the management of IBS, it is essential to select patients to be referred to the gastroenterologist, who is the figure of reference for complex cases and/or those that are difficult to manage.

It would thus be desirable to draw up shared guidelines to achieve common management paths between GPs and gastroenterologists. This could reduce unnecessary examinations and specialist referrals, consequently increasing patient satisfaction rates, and promoting a more rational allocation of healthcare spending. This could be better achieved if there were an adequate generational change in primary care.

## Figures and Tables

**Figure 1 jcm-11-03861-f001:**
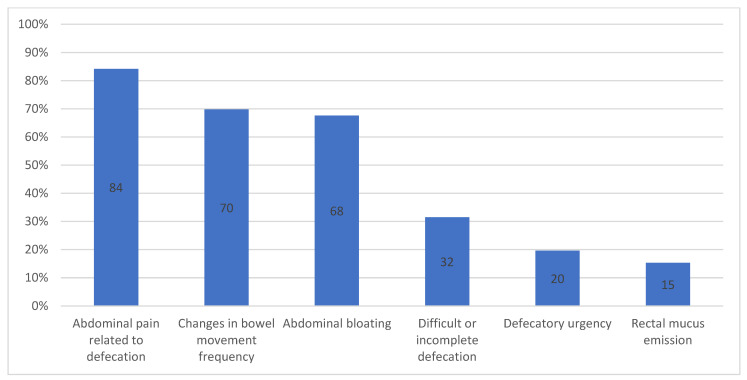
Most common symptoms used to diagnose IBS. IBS: irritable bowel syndrome.

**Figure 2 jcm-11-03861-f002:**
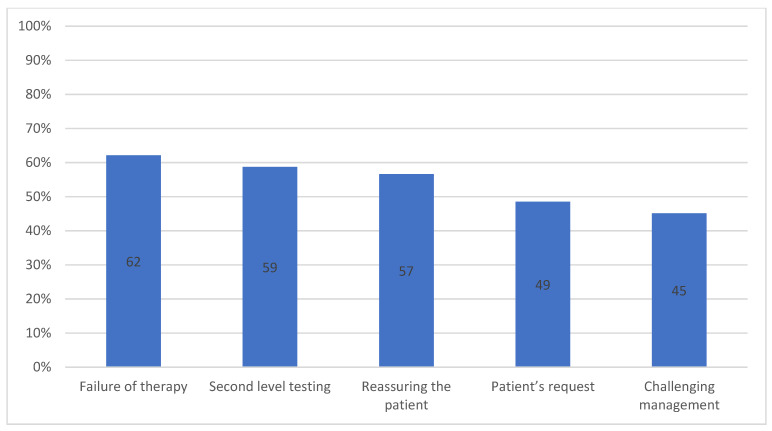
Gastroenterological consultation: frequency of different reasons.

**Table 1 jcm-11-03861-t001:** Rome IV Criteria for IBS [8].

Recurrent Abdominal Pain, on Average, at Least 1 Day/Week in the Last 3 Months, Associated with 2 or More of the Following Criteria:
Related to defecation
2. Associated with a change in frequency of stool
3. Associated with a change in form (appearance) of stool

These criteria should be fulfilled for the last 3 months with symptom onset at least 6 months before diagnosis. IBS: Irritable bowel syndrome.

**Table 2 jcm-11-03861-t002:** Main features of the 235 GPs who participated in the study.

Main Features	Results
Male/Female	139/96
Mean age ± SD (yrs.)	52.0 ± 14.6
GPs specialized in gastroenterology	28 (12.0%)
GPs considering their knowledge of IBS satisfactory	124 (52.8%)
GPs considering an update on IBS useful	92 (39.1%)

IBS: Irritable bowel syndrome.

**Table 3 jcm-11-03861-t003:** Main features of SGPs and JGPs.

Main Features	SGPs (*n* = 170)	JGPs (*n* = 65)	*p*-Value
Male/Female	118/52	21/44	0.001
Age, mean (SD), years	60 (7.6)	31.1 (4.2)	0.001
Knowledge of the RC IV (%)	81 (47.6)	47 (72.3)	0.001
Knowledge of the BS (%)	79 (46.5)	45 (69.2)	0.003

SGPs: senior general practitioners; JGPs: junior general practitioners; RC: Rome Criteria; BS: Bristol Scale.

**Table 4 jcm-11-03861-t004:** Knowledge and use of the RC IV and the BS.

Knowledge and Use	Results
Knowledge of the RC IV	128 (54.5%)
Use of RC IV in clinical practice *	97/128 (75.8%)
Approval of IBS definition of the RC IV *	112/128 (87.5%)
Willingness to add other symptoms to the IBS definition of the RC IV *	72/128 (56.3%)
Inclusion of “abdominal bloating” in the IBS definition #	61/72 (84.7%)
Inclusion of “abdominal discomfort” in the IBS definition #	31/72 (43.1%)
Knowledge of the BS	124 (52.8%)
Use of the BS in clinical practice *	92/124 (74.2%)

GPs: general practitioners; RC: Rome Criteria; BS: Bristol Scale; IBS: Irritable bowel syndrome. * Among GPs who reported being familiar with it. # Among GPs who wanted to introduce other symptoms into the IBS definition.

**Table 5 jcm-11-03861-t005:** IBS pathophysiology: frequency of “likely” responses.

“Likely” Responses	Results
Abnormal gastrointestinal motility	151 (64.2%) *
Psychological triggers	145 (61.7%)
Gut dysbiosis	135 (57.4%)
Visceral hypersensitivity	121 (51.5%)
Gastrointestinal infections	116 (49.4%)
Food intolerance and/or allergy	106 (46.3%)

IBS: Irritable bowel syndrome. * *p* < 0.05 vs. Visceral hypersensitivity, gastrointestinal infections, food intolerance and/or allergy.

## Data Availability

The data presented in this study are available on request from the corresponding author.

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
