# Peer review of "Translational Gap between Guidelines and Clinical Medicine: The Viewpoint of Italian General Practitioners in the Management of IBS"

_jcm, 2022, doi:10.3390/jcm11133861_

Round 1

Reviewer 1 Report

Authors present a study evaluating the awareness and use of Rome IV criteria and Bristol Scale for the diagnosis and management of IBS by Italian GPs. 

Authors should commended for addressing this important area of research. IBS is frequently under-looked area of research but highly prevalent in the community. Primary healthcare has the potential to improve patient outcomes through targeted education programs aimed at accurate IBS diagnosis. 

Comments: 

  1. Please provide the questionnaire used in the study in the appendix or as supplementary material
  2. Regarding questionnaire: provide the bases and evidence on how this questionnaire was developed and validated by authors
  3. How did the authors address selection bias (e.g. GPs knowledgeable to IBS are more likely to respond to study)?
  4. An important aspect, which was not addressed, is the clinical pathway for diagnosing and treating IBS (e.g. diet diary, motility study, dietary intervention, medication)
    • What is the standard clinical pathway in Italy for diagnosis and treatment of IBS?
    • Does the clinical pathway change between GPs with and without knowledge of Rome IV and BS?

The importance of evaluating GP behavior in the diagnosis and treatment of IBS should be taken into consideration. As knowledge of Rome IV/BS may not impact the clinical pathway of patient's suspected of having IBS. Hence, standardization alone may not improve clinical outcomes.    

Author Response

We thank the editor and the reviewers for their comments and suggestions, which enabled us to greatly improve our manuscript.

Here enclosed please find the “point to point” answers to comments and questions.

Authors present a study evaluating the awareness and use of Rome IV criteria and Bristol Scale for the diagnosis and management of IBS by Italian GPs. 

Authors should commended for addressing this important area of research. IBS is frequently under-looked area of research but highly prevalent in the community. Primary healthcare has the potential to improve patient outcomes through targeted education programs aimed at accurate IBS diagnosis. 

Comments: 

  1. Please provide the questionnaire used in the study in the appendix or as supplementary material.

We have done this. See supplementary material.

  1. Regarding questionnaire: provide the bases and evidence on how this questionnaire was developed and validated by authors.

As suggested by the reviewer, we have clarified this. See lines 73-75.

  1. How did the authors address selection bias (e.g. GPs knowledgeable to IBS are more likely to respond to study)?

According to the reviewer's suggestion, we have addressed this issue within the limits of the study. See lines 246-249.

  1. An important aspect, which was not addressed, is the clinical pathway for diagnosing and treating IBS (e.g. diet diary, motility study, dietary intervention, medication)
    • What is the standard clinical pathway in Italy for diagnosis and treatment of IBS?
    • Does the clinical pathway change between GPs with and without knowledge of Rome IV and BS?

The reviewer identified an important issue. However, this was not the specific aim of the study, and this issue will be addressed in a forthcoming investigation.

The importance of evaluating GP behavior in the diagnosis and treatment of IBS should be taken into consideration. As knowledge of Rome IV/BS may not impact the clinical pathway of patient's suspected of having IBS. Hence, standardization alone may not improve clinical outcomes. 

We strongly agree with the reviewer on this point, and, as stated above, we are developing a study on the diagnostic and therapeutic behaviour of GPs on IBS patients.

Reviewer 2 Report

In this work, Bellini and colleagues performed a survey to assess the awareness and the use of criteria for the diagnosis and management of IBS in an Italian group of general practitioners (GPs). The Authors included both expert and junior practitioners, from a wide geographic area. The study concluded that junior practitioners are more aware of Rome IV criteria and Bristol scale, but overall a significant proportion of GPs has an insufficient knowledge of these criteria. Moreover, the GPs propose some changes to Guidelines definition of IBS.

The study has the merit of having included a large number of GPs, which leads to interesting results and possible productive reflections. However, several points need further discussion and/or modifications, in order to allow a correct interpretation of the presented data.

Major points:

  • The Authors reported that almost half of the GPs have insufficient knowledge of IBS criteria, but about 75% of them use such criteria in clinical practice. This point involves the quality of clinical care and should be further discussed.
  • The Authors reported that “Italian GPs are … very likely anchored in their professional routine, or resistant to changes”. This appears a quite strong judgement about professionalism, and need more references or discussion. Otherwise, it should be mentioned as personal opinion of the Authors or eliminated.
  • The Authors mentioned that GP were involved from different regions. It would be useful to report more details about geographic distribution of GP that answered the survey.
  • The study lacks a control group, such as specialist gastroenterologists (that act as a specialist, and not as GP) or IBS experts. This should be made explicit and discussed.
  • IBS can be difficult to diagnose and treat, and the diagnostic algorithm also include careful exclusion of organic diseases, for example inflammatory bowel diseases or cancer. In light of the presented results, it seems even more reasonable that a gastroenterological disease should be diagnosed, treated and followed by specialist gastroenterologist, despite the involvement of GP in the process of drafting guidelines is desirable. This message should made clear in the “discussion” and “conclusion” paragraph.
  • As the Authors reported about differences between junior and expert GPs, it would be useful to report such differences within the tables.

Minor points:

  • The graphics of tables should be improved (please add title line)
  • Adding a table resuming Rome IV criteria would help the reader when interpreting the results.

Author Response

We thank the editor and the reviewers for their comments and suggestions, which enabled us to greatly improve our manuscript.

Here enclosed please find the “point to point” answers to comments and questions.

Comments and Suggestions for Authors

In this work, Bellini and colleagues performed a survey to assess the awareness and the use of criteria for the diagnosis and management of IBS in an Italian group of general practitioners (GPs). The Authors included both expert and junior practitioners, from a wide geographic area. The study concluded that junior practitioners are more aware of Rome IV criteria and Bristol scale, but overall a significant proportion of GPs has an insufficient knowledge of these criteria. Moreover, the GPs propose some changes to Guidelines definition of IBS.

The study has the merit of having included a large number of GPs, which leads to interesting results and possible productive reflections. However, several points need further discussion and/or modifications, in order to allow a correct interpretation of the presented data.

Major points:

The Authors reported that almost half of the GPs have insufficient knowledge of IBS criteria, but about 75% of them use such criteria in clinical practice. This point involves the quality of clinical care and should be further discussed.

Of the GPs, 54.5% (128) stated that they knew RC IV and of these, 75.8% (97/128) use RC IV in clinical practice. Therefore, this data does not refer to all participants in the study. Moreover, this issue is already extensively debated in the discussion. See lines 162-182.

The Authors reported that “Italian GPs are … very likely anchored in their professional routine, or resistant to changes”. This appears a quite strong judgement about professionalism, and need more references or discussion. Otherwise, it should be mentioned as personal opinion of the Authors or eliminated.

According to the reviewer's suggestion, we adjusted the text. See lines 55-61.

The Authors mentioned that GPs were involved from different regions. It would be useful to report more details about geographic distribution of GP that answered the survey.

As suggested by the reviewer we further clarified geographic distribution of the GPs enrolled. See lines 70-72.

The study lacks a control group, such as specialist gastroenterologists (that act as a specialist, and not as GP) or IBS experts. This should be made explicit and discussed.

We added this point as another limit of the study. See lines 250-252.

IBS can be difficult to diagnose and treat, and the diagnostic algorithm also include careful exclusion of organic diseases, for example inflammatory bowel diseases or cancer. In light of the presented results, it seems even more reasonable that a gastroenterological disease should be diagnosed, treated and followed by specialist gastroenterologist, despite the involvement of GP in the process of drafting guidelines is desirable. This message should made clear in the “discussion” and “conclusion” paragraph.

According to the reviewer’ suggestion we discussed this topic in the conclusions. See lines 260-264

As the Authors reported about differences between junior and expert GPs, it would be useful to report such differences within the tables.

As suggested by the reviewer, we have included a table showing the statistically significant differences between Juniors and Seniors to avoid including too many tables without significant data. See Table 3.

 Minor points:

The graphics of tables should be improved (please add title line)

We did this. See Tables 2-4-5.

Adding a table resuming Rome IV criteria would help the reader when interpreting the results.

We included Table 1.